# Contextual, Personal and Family Factors in Explaining Academic Achievement: A Multilevel Study

Carla Ortiz-de-Villate *[ID], Javier Rodríguez-Santero [ID] and Juan-Jesús Torres-Gordillo [ID]

Department of Educational Research Methods and Diagnostics, Faculty of Education Sciences, University of Seville, 41013 Seville, Spain; jarosa@us.es (J.R.-S.); juanj@us.es (J.-J.T.-G.)
* Correspondence: cortizdevillate@us.es

**Abstract:** Academic success and excellence in marks are not only due to the students' intrinsic abilities or skills. A multitude of contextual variables is involved in the teaching and learning process. This study identifies and analyses the contextual variables that most significantly affect students' academic performance. We used census data from the ESCALA tests and context questionnaires for the 2016/17 school year, both carried out by the Andalusian Agency for Educational Assessment. Hierarchical Linear Models were used in the data analysis, as they facilitate the control and description of contextual factors. In addition, differences in performance were studied according to the contextual variables that the family dimension encompasses. At all times, a contextualised cross-sectional perspective was considered, taking as a criterion the covariates with significance values lower than 0.01 in the multilevel models and contrast tests. Finally, a list of contextual variables contributing to the explanation of performance is presented. The socio-economic and cultural conditions of families, their expectations towards their children's education, and their level of involvement in schools have a significant influence on the academic success of primary school students. Academic success is higher among students with families who play an active role and have positive expectations regarding learning.

**Keywords:** academic achievement; factors; family; educational improvement; educational efficiency

## 1. Introduction

This study is part of the School Effectiveness and School Improvement Movement (SESI), a theoretical–practical research approach with the purpose of understanding and explaining the reality of education [1]. It is based on two different but complementary lines of research, which share the aim of understanding the factors involved in learning and academic success [2]. First, there is the theoretical side, of a quantitative nature, which focuses on the study of school effectiveness in terms of performance. Second, there is the practical aspect, of a qualitative nature, aimed at school improvement. In addition to its relevance today, the school effectiveness and improvement movement has a long history in the field of educational research, originating in 1966 with the Coleman Report, which identified the shortcomings that existed in relation to school performance and educational transformation [1].

Numerous research studies have been carried out in this respect, such as Weber's study [3] for the variable analysis of processual and contextual typology or the contributions of Aitkin and Longford [4], which consolidated the foundations of the field of research using multilevel statistical techniques. Also, the studies by Reynolds, Hopkins, and Stoll [5] and those carried out by Creemers and Reezigt [6] are precursors of the school improvement and effectiveness movement. However, the International Handbook of School Effectiveness and Improvement by Townsend and Avalos [7] is considered the most outstanding and noteworthy work in this line of research.

Over the years, educational research on school effectiveness and improvement has provided valuable insights in explaining the functioning of schools and the validity of their

actions, as well as the contextual and personal factors that affect students' academic performance. Some studies have even claimed that the practices of the most effective schools can significantly increase the academic achievement of minority students or students with greater learning difficulties [8]. For this reason, much of the research in the field has focused on studying the factors that characterise schools and students in terms of achievement.

As shown in the work of García-Jiménez et al. [9], Martínez-Abad et al. [10], Luzarraga et al. [11], and Jayanthi et al. [12], the socio-economic and cultural conditions of families, family characteristics, gender, and geographical origin have a considerable influence on learning. In addition, achievement beliefs and expectations [13,14], resilience to cope with conflict situations [8], and place of residence or social group [15] are factors related to students that have been studied for their strong influence on academic success. Another of the relevant variables for performance is the time students spend doing extracurricular activities or viewing content on digital screens [16], as they switch between places and times to carry out work or learning activities [17]. Therefore, academic success is not exclusively due to students' intrinsic abilities and skills. The contextual or psychosocial particularities of the students, the characteristics of the school, and the agents involved can facilitate learning or, on the contrary, become a risk factor for academic achievement [15]. The study of school effectiveness must have a shared vision between excellence and equity [18] and control for the variant effect of those contextual factors that characterise students and schools [19,20].

The family environment, as the first socialising agent of the individual, is also a major influence on learning [21], especially in the early stages of education [22]. Young people interact continuously with their relatives and internalise constructs, attitudes, and behavioural patterns for their social performance in the closest environments [23]. The family institution is a socio-cultural frame of reference that contributes to the transmission and assimilation of values, thoughts and expectations towards the teaching and learning process. For this reason, there must be cooperation and collaboration between both sectors which is unavoidable in order to establish optimal conditions for student development and to facilitate the achievement of academic objectives [24–27].

The scientific literature reflects a growing presence of studies linked to the study of the family dimension and its effect on academic performance, for example, [28–32]. The study by Benner et al. [33] shows that the involvement and participation of families in the school context contributes to the development of favourable expectations and increases the levels of perceived self-efficacy in students. Häfner et al. [34] also report this, provided that families are characterised by a high level of interest in learning and high expectations towards their children's education. However, the percentage of families who participate in school tasks such as volunteering, or community workshops is very low, and it is usually mothers who are more actively involved in such practices [35]. Other studies suggest that the socio-economic and cultural conditions of families, parents' occupation, and expectations towards learning are the family factors with the greatest weight in explaining achievement [36–39]. The works of Froiland and Davinson [40] and Touron et al. [41] show that parental emotional support, the social welfare of families and their socio-economic conditions are linked to better academic achievement. In their study on the impact of families on achievement, Castro et al. [13] also found a strong correlation between parents' educational expectations and their children's academic achievement. Specifically, high family expectations are associated with students achieving better grades in school.

Regarding the socio-economic and cultural conditions of families, Barnardi and Raquena [42] found differences in achievement according to social background or class. The results of the study showed that, unlike upper-class pupils, those from lower-middle socio-economic backgrounds have more difficulties in learning. As these authors suggest, the worst-off pupils assumed their disadvantages and were more concerned about their families and less motivated towards their learning [30]. In addition, it has been shown that families with a higher level of income are more involved in their children's development, although the relationship is inverse when it comes to school participation [35].

Similar results were obtained in the work of Ali et al. [43], Davis-Kean [44] and Harju-Luukkainen et al. [45], with students from rich socio-economic and cultural backgrounds ultimately achieving better academic performance. Consequently, the combined effect of being poor and belonging to ethnic minorities or groups at risk of exclusion increases low educational outcomes [28] and limits students' resilience [8].

The importance of these problems requires large-scale studies that analyse and consider the effect of contextual and personal factors on learning. Findings in this field of research can be useful for the design of improvement programmes and actions that favour equal opportunities and increase students' academic success.

## 2. Materials and Methods

The study presented is a secondary analysis based on the results obtained in the edition of the ESCALA assessment test corresponding to the 2016/17 school year. The ESCALA tests were carried out by the Andalusian Agency for Educational Assessment (AGAEVE), to ascertain the level of learning achieved by Andalusian pupils in the 3rd year of primary school (PS) in the instrumental subjects of mathematical reasoning and linguistic communication. These tests, from their first edition in the 2010/11 school year until the last one in the 2018/19 school year, were conducted annually to provide objective and valuable information about the reading, writing, and numeracy skills and abilities of students in the first few years of compulsory schooling. Attached to these tests were context questionnaires to be completed by families. These questionnaires collected personal, socio-cultural and economic data that characterise the students and families of the schools in order to explore their relationship with the levels of academic performance. However, few studies have looked at how these tests could help to make significant progress in the field of research on school effectiveness and school improvement.

Taking into consideration the scope of the project underlying this work, it has been possible to access census data obtained in the ESCALA tests, corresponding to the 2016/17 school year. Thus, an ex post facto, quantitative, cross-sectional, contextualised, and correlational study has been proposed to detect covariates that affect academic performance. In addition, in previous studies it was hypothesised that academic success in the early stages of schooling differs according to the socio-cultural characteristics of families, their involvement in school and their expectations of achievement. To further elaborate on this premise, differences in performance were studied according to the contextual variables that the family dimension encompasses.

### 2.1. Participants

The selection of participants for the study was carried out on the basis of the schools and students who participated in the most recent edition of the ESCALA tests made available by the Ministry of Education of the Andalusia Regional Government. For this reason, the data that were analysed to correspond to the Diagnostic Assessments (DA), provided by AGAEVE, which were completed by pupils in the 3rd year of PS in Andalusia in the 2016/17 school year. We worked with census data, considering all students and the diversity of schools that make up the Andalusian Education System. In order to offer a response as close as possible to the educational reality, we considered the dual or triad system in which public, private, and subsidised schools coexist, and we simultaneously analysed the data of the schools and students belonging to the different types of schools.

Originally, the participating sample amounted to 2525 schools, with 94,092 students enrolled in the 3rd year of PS. The large amount of data to be studied made it necessary to adjust the sample size based on the constituent size of the schools included in the study. Schools with fewer than 12 respondents for DA in the 3rd year of primary school were discarded. Students who had not participated in the DA or who did not have the respective family context questionnaires were also discarded. Finally, the sample participating in the study was reduced to 79,806 students, enrolled in 2092 schools. Most of the schools are public (76.4%) and, to a lesser extent, there are charter schools (21%) and private schools



(2.6%). In terms of pupils, 50.9% are boys and the remaining 49.1% are girls while 98% of the students were born in 2009, so their age was around 8 years old at the time the DA were applied. In addition, the contextual information required in the context questionnaires for families was used.

### 2.2. Instruments and Variables

The instruments used to collect the data correspond to the achievement tests and context questionnaires attached to these assessments. These tests previously collected students' results in numeracy and language and the contextual factors that characterised them. All the information collected during the implementation of the DA in the 2016/17 school year was used as a database for this work. Specifically, the scores obtained by students in mathematical reasoning and linguistic communication skills were used as dependent variables in the multilevel models and independent variables in the differential study. The family context questionnaires helped to specify the covariates relevant to the research. The latter refer to the personal and contextual characteristics of students and families in the schools that, in some way, help to explain the level of academic performance achieved. The covariates incorporated in the Hierarchical Linear Models (HLM) were purely contextual in nature. In other words, they are factors that cannot be controlled or affected by schools.

The covariates that were incorporated in the models for Level 1 are purely contextual factors that characterise students. The covariates included in the Level 2 analysis focus on aspects characteristic of schools. The latter were aggregated from the covariates that made up Level 1 to estimate indices for each school as proportions and averages of student achievement test scores. In the next lines of research, the covariates incorporated into the HLM are presented, with (L1) the factors describing the students and (L2) the variables contextualising the schools.

- Gender

**(L1)** Gender of the students, with a value of 0 for the condition of being a boy and a value of 1 for girls.
**(L2)** Total number of schools that operate under a gender-segregated grouping model.

- ISEC

**(L1)** Socio-economic and Cultural Index of the families. This is obtained from the factor analysis of the relevant indicators in the context questionnaires, such as parents' level of education, professional category, or material resources available in the household. The values are expressed as a normalised factor score.
**(L2)** Average Socio-Economic and Cultural Index of the student body of the school.

- Family–school involvement

**(L1)** Degree of involvement and participation of families in the school.
**(L2)** Average involvement of the students' families in school activities and dynamics.

- Family–study involvement

**(L1)** Degree of involvement of families in their children's study and learning.
**(L2)** Average involvement of the students' families in study and learning.

- Family expectations

**(L1)** Level of families' expectations of their children's education and learning.
**(L2)** Average level of families' expectations of the school.

- Reader engagement

**(L1)** Degree of responsibility of students and families for reading. This is obtained as the sum of the indicators linked to the construct.
**(L2)** Average level of responsibility of students and families in the school for reading.

- Bedtime

**(L1)** Time at which students go to bed during the school term.
**(L2)** Average bedtime of students at the school.

- Extracurricular activities

**(L1)** Number and type of extracurricular activities in which students are enrolled.
**(L2)** Average number of extracurricular activities in which the school's students are enrolled.

- Screen time

**(L1)** Time, in hours, that students spend on activities involving digital screens.
**(L2)** Average time spent by students at the school using digital screens.

- Amount of homework

**(L1)** Volume of homework students have to do at home.
**(L2)** Rate of students in the school with homework.

- Time spent on homework

**(L1)** Time, in minutes, spent by students doing homework.
**(L2)** Average amount of time school students spend on homework at home.

- Type of school

**(L2)** School network, including public, private, and state-subsidised schools.

*2.3. Analysis*

The data recorded in the assessment tests and context questionnaires present a hierarchical structure in two ranks or stages (students and schools). For this purpose, and, to identify and study the impact of contextual variables on academic performance, multilevel statistical techniques were used. HLM in particular were used as they facilitate the study of variance when predictor variables are arranged in two or more levels of nesting. Moreover, their application allows for the control and description of contextual factors for students and the schools in which they are enrolled. In the model run, the fixed or null effects equation was used as a reference, according to which the distribution for the random effects models was estimated. In the model, students' scores in mathematical reasoning and linguistic communication were included as dependent variables at Level 1. Then, the effect of covariates at both nesting levels was retained, consolidating a valid multilevel model to determine the contextual factors that most significantly affect learning.

The multiple linear regression equation of the model for Level 1 shows the variation of the student's score for each criterion variable:

$$\text{Structural part}: Y_{ij} = \beta_{0j} + \sum_{q=1}^{Q} \beta_{qj} X_{qij} + r_{ij}$$

$$\text{Probabilistic part}: r_{ij} \sim N(O, \sigma^2)$$

where $Y_{ij}$ is the score of student $i$ of school $j$ in the competencies assessed. $\beta_{0j}$ is the average score of the school in each competency assessed. $\beta_{qj}$ is the linear effect of the covariate $X_q$ of the students. $X_{qij}$ is the score of pupil $i$ of school $j$ for covariate $X_q$. Lastly, $r_{ij}$ corresponds to the difference between the obtained score and the expected score of student $i$ in school $j$ (residual).

The linear equation of the model for Level 2 indicates the influence of the covariates that refer to schools for each competency assessed. The factors for the school level were aggregated or constituted as fixed effect covariates on the basis of Level 1.

$$\text{Structural part}: \beta_{0j} = \gamma_{00} + \sum_{s=1}^{S} \gamma_{0s} W_{sj} + u_{0j}$$

$$\text{Probabilistic part}: u_{0j} \sim N(O, \tau_{00})$$

where $B_{0j}$ corresponds to the average scores for each school $j$. $Y_{00}$ is the proportional effect present in schools. $Y_{0s}$ is the linear influence of the covariate $W_s$ on average school

performance $j$. $W_{sj}$ is the value of school $j$ in covariate $W_s$. $U_{0j}$ is the residual difference between schools.

The equations that constitute each level of analysis are combined or grouped in a mixed model that corresponds to the following equation:

$$Y_{ij} = \gamma_{00} + \sum_{s=1}^{S}\gamma_{0s}\,W_{sj} + \sum_{q=1}^{Q}\beta_{qj}\,X_{qij} + u_{0j} + r_{ij}$$

From the mixed-effects model, multilevel or hierarchical models emerge.

Two models were run, one for each competency assessed, and a list of covariates contributing to the explanation of performance was obtained.

Finally, a differential analysis was conducted to study the differences in performance as a function of the contextual variables that the family dimension encompasses. Considering the sample size, normality of the scores was assumed (central limit theorem) and parametric contrast tests were applied. Specifically, the '$t$' test and Anova were used for the comparison of means, both for independent samples. The means obtained by each group of students were compared according to the level of achievement expectations of the families (0 = low; 1 = high), the degree of involvement with learning (0 = low; 1 = high), the level of involvement with the school (0 = low; 1 = high) and the socio-economic level (0 = low; 1 = medium; 2 = high). Scores in mathematical reasoning and linguistic communication were included as independent variables. In both tests, the baseline hypothesis assumes that the differences or effect of interest is zero. The alternative hypothesis assumes the existence of statistically significant differences in performance between the groups considered. The effect of size was estimated by applying Cohen's d. At all times, a contextualised cross-sectional perspective was considered, taking as a criterion the covariates with significance values of less than 0.01 in the multilevel models and contrast tests for the competences assessed.

## 3. Results

After the analysis was carried out, two multilevel models were obtained, one for each competency or subject assessed in the achievement tests. Only covariates with a significant significance value ($p < 0.01$) were considered; that is, those personal and contextual variables with a degree of association with or influence on learning that is highly representative and applicable in explaining performance. This criterion was maintained for covariates at the student (L1) and school (L2) levels.

### 3.1. Contextual Variables Associated with Academic Performance

The relevant factors for both models are determined based on the value 0. From this, the influence they exert on the scores is calculated according to the estimated values presented in the models for mathematical reasoning and linguistic communication (Table 1).

According to the intercept values, the average scores of male students are 279.86 in mathematical reasoning and 254.51 in linguistic communication. This is the central point for the explanation of these models since the rest of the covariates are initially considered null. Taking into account the value of the intersection, for each point increase in the covariates, the estimated amount corresponding to each competence assessed is added or subtracted. Therefore, girls obtain somewhat higher mathematics scores (282.89) than boys, as 3.03 points (estimated value for gender) are added to the final performance. Even more remarkable are the gender differences in language proficiency. In this case, being a girl means achieving considerably better results than boys (282.79). The gender variable stands out as a significant factor influencing the academic performance of primary school students, with girls achieving the best results.

**Table 1.** Hierarchical Linear Models for mathematical reasoning and linguistic communication.

| Covariates [1] | Mathematical Reasoning | | Linguistic Communication | |
|---|---|---|---|---|
| | Estimate | Error | Estimate | Error |
| Intercept | 279.86093 | 23.441 | 254.51318 | 23.73050 |
| Gender(L1) | 3.03436 | 0.598 | 28.28102 | 0.59004 |
| Bedtime(L1) | 2.87843 | 0.422 | 2.23227 | 0.41568 |
| Number homework(L1) | −2.56636 | 0.488 | −2.55005 | 0.48049 |
| Homework time(L1) | −16.50689 | 0.365 | −17.43667 | 0.36180 |
| Reader engagement(L1) | 0.51829 | 0.117 | 0.69994 | 0.11570 |
| ISEC(L1) | 15.33066 | 0.387 | 14.78731 | 0.38493 |
| Extracurriculars(L1) | 5.34834 | 0.365 | 5.08663 | 0.36016 |
| Screen time(L1) | 1.62210 | 0.254 | 0.93771 | 0.25078 |
| Family expectations(L1) | 4.71285 | 0.122 | 4.74462 | 0.12091 |
| Involvement F-L(L1) | −0.97194 | 0.157 | −0.44312 | 0.15503 |
| Implicación F-E(L1) | 0.77559 | 0.099 | 0.60571 | 0.09783 |
| Bedtime(L2) | 12.11171 | 4.618 | 14.28253 | 4.75651 |
| Amount homework(L2) | 26.41486 | 5.819 | 23.90230 | 5.92461 |
| Homework time(L2) | 16.87234 | 3.211 | 21.13964 | 3.32228 |
| Reader engagement(L2) | 5.01907 | 1.469 | 5.05168 | 1.62781 |
| Involvement F-S(L2) | 2.11027 | 0.785 | 2.73122 | 0.773370 |
| Gender(L2) | - | - | −24.99780 | 8.85128 |
| ISEC(L2) | - | - | 10.96174 | 2.49378 |

[1] Covariates with significance values lower than 0.01.

Another important variable in the explanation of performance is the time students spend on homework. In mathematics proficiency, there is a negative estimate value (−16.5) on the covariate. In other words, students who spend more time doing homework obtain much lower scores in mathematics (263.35). The results are similar for the model referring to language proficiency. In this case, students who spend more time doing homework are the ones who ultimately perform lower (−17.4) in language. This situation probably indicates that, at this age, those who spend more time doing homework are those students who have more learning difficulties. At the school level (n2), homework time also shows considerable estimation values. However, the sign of the values for both models is reversed and turns out to be in favour of learning. It means that a higher average time spent by school students on homework improves performance in mathematics (16.87) and language (21.13). Also, the time students go to bed on weekdays during the school term has a significant influence on performance in mathematical reasoning (2.87) and linguistic communication (2.23), especially at the school level (L2), where the figures increase and are around 12 and 14 points. The estimation values for the covariate are positive and report that students who go to bed later than 10 p.m. achieve better grades. Taking into account the positive effects of after-school activities on learning (5 points more), it is possible that students who go to bed later have more complete routines and do more after-school activities in their leisure time.

*3.2. The Family Dimension in Explaining Academic Performance*

The contextual variables that refer directly to the family dimension also have a considerable impact on academic performance. As shown in the results in Table 1, for each point increase in the ISEC variable, mathematics and language scores increase by 15.33 and 14.78 points respectively. Therefore, the group of students whose families have a higher socio-economic and cultural index achieve better results in the competencies assessed. Other relevant covariates that influence student learning are family expectations, family participation in the teaching and learning process, and family involvement with the school. In all of them, the estimated values for mathematics and language are in favour of achievement. High family expectations are associated with better academic results, as each point increase in the covariate adds 4.7 points in both competences. However, their participation and involvement with the school show low estimate values (0.77; 0.60) but increase for

Level 2 schools. That is, higher average involvement of school families in school activities and dynamics results in better outcomes for students.

The results of the 't' test (Table 2), for the set of covariates analysed, show significance values of less than 0.01. Thus, it is assumed, with a confidence level of 99%, that there are differences in the students' performance according to the contextual variables included in the family dimension.

**Table 2.** Results of the t test for family covariates.

| Covariates [1] | Competence | Sig. | Difference | d |
|---|---|---|---|---|
| Family expectations | Mathematical reasoning | 0.000 | 35.466 | 0.38 |
| | Linguistic communication | 0.000 | 37.030 | 0.4 |
| Involvement F-L | Mathematical reasoning | 0.000 | 11.730 | 0.13 |
| | Linguistic communication | 0.000 | 15.693 | 0.18 |
| Involvement F-S | Mathematical reasoning | 0.000 | 14.580 | 0.15 |
| | Linguistic communication | 0.000 | 15.245 | 0.17 |

[1] Covariates with significance values lower than 0.01.

The mean difference for the covariate family expectations is considerable, both in mathematics (35.46) and language (37.03). Taking into account the value of and evidence provided by the difference in means, it is affirmed that the group of students whose families have high expectations towards their learning perform better in the subjects assessed than students whose families have lower expectations. The magnitude of the differences in this case could be considered moderate (d = 0.38; 0.4). Students whose families are involved and engaged with the school and their learning also score higher in mathematical reasoning and linguistic communication than students with less involved families. However, the magnitude of the differences could be considered small (d = is around 0.2).

At the same time, differences in performance were studied based on the ISEC covariate and its different categories (high, medium and low). The results of the Anova test (Table 3) showed significant significance values ($p < 0.001$), affirming the existence of differences in students' learning according to the socio-economic and cultural conditions of their families. According to the data presented in Table 3, the group of students whose families are characterised by a high ISEC obtain better scores in language (77.918 mean points higher) and mathematics (71.514 mean points higher) than students with a low ISEC. Similar results are observed for students with a high and medium ISEC, with better scores for the high ISEC group by 35.654 and 32.999 mean points depending on the proficiency tested. Students whose families have a low socio-economic and cultural background perform worse compared to those with a high or medium ISEC. In short, it can be said that there are differences in student learning depending on the ISEC of the families, with students with a high ISEC achieving better performance in mathematical reasoning and linguistic communication. The magnitude of the differences could be considered moderate (d values are close to 0.5).

**Table 3.** Scheffé test results.

| Competence | ISEC | Sig. [1] | Difference | d |
|---|---|---|---|---|
| Mathematical reasoning | High ISEC/Medium ISEC | 0.000 | 32.999 | 0.37 |
| | High ISEC/Low ISEC | 0.000 | 71.514 | 0.57 |
| | Medium ISEC/Low ISEC | 0.000 | 38.515 | 0.39 |
| Linguistic communication | High ISEC/Medium ISEC | 0.000 | 35.654 | 0.4 |
| | High ISEC/Low ISEC | 0.000 | 77.918 | 0.62 |
| | Medium ISEC/Low ISEC | 0.000 | 42.264 | 0.43 |

[1] Covariates with significance values lower than 0.01.

## 4. Discussion

The involvement of families in the teaching and learning process has been widely studied to explain the academic success of students during the early stages of education [8,13,14,28,32,33]. However, few studies have focused on the Spanish context to determine which family or contextual factors have an impact on academic performance [9,21,41,42] and only a small part of these studies considers the multilevel effect of these factors in the educational environment [1,20].

In this study, multilevel statistical techniques were used to detect and assess the effect of personal and contextual variables that characterise students and schools in terms of levels of academic achievement. The results of the multilevel study show that variables such as the gender of students, the amount of time spent on homework, bedtime, and after-school activities have a significant influence on the academic achievement of Andalusian primary school students. In the early educational stages, girls perform better academically in grammar, reading, and numeracy than boys. These results may support arguments that girls have earlier brain development for acquiring reading and writing skills or have a better ability to concentrate. In addition, students who spend more hours doing homework are those who perform less well in the skills tested. This probably indicates that, at the age of 8, those who spend more time doing homework are those students who have more learning difficulties. Family factors such as socio-economic and cultural conditions, their expectations of achievement, and their level of participation and involvement with schools have a significant impact on academic performance. The results show that the students whose families are more involved, have high expectations of achievement and have higher incomes, achieve better results in mathematical reasoning and linguistic communication. However, the weight of the variables related to family participation and involvement in school is not as high as the effect of belonging to a family with a high-income level.

In short, there is a great variability in terms of the contextual variables that affect achievement, but these are repeated for mathematical reasoning and linguistic communication skills. Therefore, academic success and excellence in scores are not only due to students' intrinsic abilities or skills. A multitude of contextual variables is involved in the teaching and learning process and mere scores do not take into account the variant effect of these factors. As found in the work of Lizasoain et al. [18] and Hernández-Castilla et al. [46], academic effectiveness and achievement must be determined by excellence and equity to compensate for the inequalities that arise in school environments.

The results of the differential study turned out to be in line with the approaches that the scientific literature takes regarding the family dimension and its link to students' academic performance [10,11,33,37]. Significant differences in learning have been found depending on the socio-economic and cultural index of families, their expectations towards education and, to a lesser extent, their involvement and participation in school and learning. Low socio-economic status is a risk factor for learning, as the results show that students whose families are characterised by a low ISEC, score lower in mathematics and language than students with a higher socio-economic status. Additionally, the group of students whose families establish a close bond with schools and have high expectations for learning achieve higher scores in the competencies assessed. These results are in line with Rodríguez-Santero and Gil-Flores [47], Froiland and Davinson [40], Tourón et al. [41] and Davis-Kean [44], who have highlighted the support and involvement of families with the school and their children's education as key factors in academic achievement. In line with these findings, recent studies in the USA and China have shown that the quality and quantity of family involvement in reading at home is associated with children's outcomes in mathematics and language skills [48–50], while socio-economic conditions, ethnicity and children's age affect the quality and quantity of family involvement in reading.

Unfortunately, it is useless to address class differences or the socio-economic conditions of families, as we are part of a stratified society that has been in place for years. However, schools can compensate for inequalities by taking action to ensure that educational and academic resources are made available to students with the greatest needs.



As explained by Bao et al. [51] the attention of families, especially mothers, to literacy in low-income households is key to the success and motivation of students in the early stages of education. Therefore, it is recommended that schools promote specific psycho-emotional programmes or formulate policies that promote home-based strategies to involve families in students' learning and literacy. In addition, the implementation of a dialogic model, inclusive groupings, or participatory methodologies for the family and the school could be a very useful option to foster a sense of belonging, of collaboration, and of educational support among students and families.

## 5. Conclusions and Limitations

This section mentions the most relevant educational findings of the study, as well as the limitations and future implications for further studies of educational effectiveness and improvement.

First and foremost, it is worth highlighting the results of analyses suggesting that the gender of students, the amount of time spent on homework, bedtime, and extracurricular activities have a significant influence on the academic achievement of primary school students in Andalusia. Significant differences in performance have also been found according to the socio-economic and cultural index of families, their expectations towards education and, to a lesser extent, their involvement and participation in school and learning. Low socio-economic status is a risk factor for learning, and it is suggested that school efforts should be balanced and complemented by promoting home-based strategies to involve families in their children's learning and literacy.

In short, the use of hierarchical linear models has made it possible to detect contextual factors that contribute to explaining performance in a way that is completely valid and applicable to the field of education [52]. Moreover, its multilevel effect has facilitated the work of control and adjustment for the covariates that affect academic performance at the student and school levels. Martínez-Abad et al. [10] asserts that, in the study of school effectiveness and improvement, HLM are the only statistical techniques that, so far, offer a faithful response to the complex reality of learning and education as a construct.

This work adds to the academic debate on the effectiveness and school improvement movement and contributes to the dialogue and reflection on the factors that affect the academic achievement of students in primary school. In particular, the results obtained have great implications for the theoretical and practical design of school improvement actions in the Andalusian context, which can be generalised to other territories with similar characteristics in practice. Although the study of educational reality is complex and changing, the size of the sample and the results of other associated studies on the Spanish population that show similarities in terms of the variables involved in academic performance [1,9,20,21,42], reinforce the possibilities of integration and extension to other territories.

Some of the limitations of this work are to be found in the list of covariates incorporated in the multilevel models, since they were based on the data that AGAEVE had collected in the achievement tests for the 2016/17 school year. Therefore, it has not been possible to consider the influence of other relevant factors, such as immigrant status or the number of students repeating academic years. In addition, some sincerity biases can be found in the data as a result of the context questionnaires that were filled in by the families. Even so, this work provides important findings for the field of school effectiveness and improvement and the study of the role of the family in the academic success of students. Finally, regarding future research, a more specific study incorporating a larger number of associated variables or adopting a qualitative perspective is proposed in order to examine in greater depth the factors that affect students' academic performance.

**Author Contributions:** Conceptualization and design of research framework, C.O.-d.-V., J.R.-S. and J.-J.T.-G.; methodology, C.O.-d.-V., J.R.-S. and J.-J.T.-G.; software, C.O.-d.-V., J.R.-S. and J.-J.T.-G.; validation, C.O.-d.-V., J.R.-S. and J.-J.T.-G.; formal analysis, C.O.-d.-V., J.R.-S. and J.-J.T.-G.; data curation, C.O.-d.-V., J.R.-S. and J.-J.T.-G.; writing—original draft preparation, C.O.-d.-V.; writing—review and editing, C.O.-d.-V., J.R.-S. and J.-J.T.-G.; project administration, J.R.-S. and J.-J.T.-G.; funding acquisition, J.R.-S. and J.-J.T.-G. All authors have read and agreed to the published version of the manuscript.

**Funding:** Project EDU2017-84649-P funded by MCIN/AEI/10.13039/501100011033 and FEDER "A way to make Europe".

**Institutional Review Board Statement:** Not applicable.

**Informed Consent Statement:** Not applicable.

**Acknowledgments:** Ministry of Science, Innovation and Universities of Spain and University of Seville.

**Conflicts of Interest:** The authors declare no conflict of interest.

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
