# Peer review of "Contextual, Personal and Family Factors in Explaining Academic Achievement: A Multilevel Study"

_sustainability, doi:10.3390/su132011297_

Round 1
Reviewer 1 Report
The manuscript is devoted to assessing the factors affecting primary school students’ performance. The analysis was based on the census data referring to the 2016-17 school year in Andalusian region, by using the Hierarchical Linear Model. Results show that socio-economic and cultural factors significantly affect the students’ academic success.
The topic is within the scope of the journal, and it is surely interesting in the field of the socio-cultural assessment of childhood education and learning capabilities. However, I believe that significant improvements are needed because the analysis lacks some crucial points. Specifically, please consider the following questions and suggestions:
- The Introduction section needs to be improved, by extending the analysis of the state-of-the-art related to the considered approach. Please consider the following suggested references:
- Agyeman G.A., Frimpong E.A., Ganyo E.R. (2016). Students’ perception of socio-cultural factors affecting academic performance. American Academic Scientific Research Journal for Engineering, Technology, and Sciences, 19(1)
- Borman G.D., Rchuba L.T. (2001). Academic Success among Poor and Minority Students: An Analysis of Competing Models of School Effects.
- Sad S.N., Oguz G. (2013). Primary school students’ parents’ level of involvement into their children’s education. Education Sciences: theory and practice, 13(2), 1006-1011.
- The Materials and Methods section should provide more details about the criteria considered for selecting the students’ samples. What about extending the analysis to further school years and pupils of other years of primary school (e.g. 2nd and 4th-year students)?
- Further details of the analytic approach composing the HLM should be provided.
- How do you think that the results may be extended to the overall Spanish national area?
- How do you manage the biases related to the variable level of reliability due to the families’ responses to the questionnaire?
- T-test and Anova statistics should be better introduced in the text, by briefly introducing the analytic approach applied to their application.
- Some statistical graphs may be included to better assess the observed results at the different levels of the analysis.
- A Conclusions section should be added, including the overall results of the analysis, its limitations and providing further information on the next step of the research. They are only mentioned in the Discussion section.
- An overall check of English may improve the reading.
Kind regards
Author Response
The response is uploaded as a Word file.

Reviewer 2 Report
This is a fairly clean paper. I am attaching my few comments - and they are very few and involve small points.

Author Response

(The authors gave the same response as above.)

Round 2
Reviewer 1 Report
The authors have addressed the reviewers' comments, by extensively improving the manuscript. I believe that it is suitable for acceptance in the current version.
Kind regards,